

# 1 Ice core chemistry database: an Antarctic compilation of
# 2 sodium and sulphate records spanning the past 2000 years.

Elizabeth R. Thomas[1], Diana O. Vladimirova[1], Dieter Tetzner[1], B. Daniel Emanuelsson[1],
Nathan Chellman[2], Daniel A. Dixon[3], Hugues Goosse[4], Mackenzie M. Grieman[5], Amy C.F.
King[1], Michael Sigl[6], Danielle G Udy[7], Tessa R. Vance[8], Dominic A. Winski[3], V. Holly L.
Winton[9], Nancy A.N. Bertler[9,10], Akira Hori[11], Chavarukonam.M Laluraj[12], Joseph R.
McConnell[2], Yuko Motizuki[13], Kazuya Takahashi[13], Hideaki Motoyama[14], Yoichi Nakai[13],
Franciele Schwanck[15], Jefferson Cardia Simões[15], Filipe G. L. Lindau[15], Mirko Severi[16], Rita
Traversi[16], Sarah Wauthy[17], Cunde Xiao[18], Jiao Yang[19], Ellen Mosely-Thompson[20], Tamara
V. Khodzher[21], Ludmila P. Golobokova[21], Alexey A. Ekaykin[22]
[1]Ice Dynamics and Paleoclimate, British Antarctic Survey, High Cross, Madingley Road, Cambridge,
CB3 0ET, UK
[2]Division of Hydrologic Sciences, Desert Research Institute, Reno, NV, 89512, USA
[3]Climate Change Institute, University of Maine, 5790 Bryand Global Science Center, Orono, ME,
04469, USA.
[4]Earth and Life Institute, Universite catholique de Louvain, Place Pasteur 3, 1348 Louvain-la-Neuve,
Belgium
[5]Department of Chemistry, Reed College, 3203 Woodstock Blvd., Portland, Oregon, 97202, USA
[6]Climate and Environmental Physics (CEP), Physics Institute & Oeschger Centre for Climate Change
Research (OCCR), University of Bern, Sidlerstrasse 5, 3012 Bern, Switzerland
[7]Institute for Marine and Antarctic Studies, University of Tasmania, 20 Castray Esplanade, Battery
Point TAS 7004, Australia
[8]Australian Antarctic Program Partnership, Institute for Marine & Antarctic Studies, University of
Tasmania, Hobart, Australia
[9]Antarctic Research Centre, Victoria University of Wellington, Kelburn Parade, Kelburn, Wellington
6021, New Zealand
[10]National Ice Core Facility, GNS Science, 30 Gracefield Rd, Gracefield 5040, New Zealand
[11]Kitami Institute of Technology, 090-8507, Japan
[12]National Centre for Polar and Ocean Research (NCPOR), Ministry of Earth Sciences, Vasco-da
Gama, Goa 403804, India
[13]RIKEN Nishina Center for Accelerator-Based Science, 2-1 Hirosawa, Wako, Saitama 351-0198,
Japan
[14]National Institute of Polar Research, Tachikawa, Tokyo 190-8518, Japan
[15]Centro Polar e Climático, Universidade Federal do Rio Grande do Sul, Porto Alegre, 91501-970,
Brazil
[16]Chemistry Dept. "Ugo Schiff", University of Florence, 50019, Sesto F.no, Florence, Italy.



[17]Laboratoire de Glaciologie, Department Geosciences, Environnement et Societe, Universite Libre de
Bruxelles, 1050 Brussels, Belgium
[18]State Key Laboratory of Earth Surface Processes and Resource Ecology, Beijing Normal University,
China
[19]State Key Laboratory of Cryospheric Science, Northwest Institute of Eco-Environment and
Resources, Chinese Academy of Sciences, Lanzhou 730000, China
[20]Byrd Polar and Climate Research Center, The Ohio State University, 1090 Carmack Rd. Columbus
OH 43210 USA
[21] Limnological Institute of Siberian Branch of the Russian Academy of Sciences), Irkutsk, 664033,
Russia
[22] Arctic and Antarctic Research Institute), 38 Bering st, St Petersburg, 199397, Russia
*Correspondence to:* Elizabeth R. Thomas (lith@bas.ac.uk)





**Abstract.** Changes in sea ice conditions and atmospheric circulation over the Southern Ocean play an important
role in modulating Antarctic climate. However, observations of both sea ice and wind conditions are limited in
Antarctica and the Southern Ocean, both temporally and spatially, prior to the satellite era (1970 onwards). Ice
core chemistry data can be used to reconstruct changes over annual, decadal, and millennial timescales. To
facilitate sea ice and wind reconstructions, the CLIVASH2k working group has compiled a database of two
species, sodium [Na$^+$] and sulphate [SO$_4^{2-}$], commonly measured ionic species. The database contains records
from 105 Antarctic ice cores, containing records with a maximum age duration of 2000 years. An initial filter
has been applied, based on evaluation against climate observations, to identify sites suitable for reconstructing
past sea ice conditions, wind strength, or atmospheric circulation.


## 1 Introduction

Changes in wind strength and atmospheric circulation, notably the positive phase of the Southern Annular Mode
(SAM), have been related to increased Antarctic snowfall (Thomas et al., 2017; Thomas et al., 2008; Medley
and Thomas, 2019) and the widespread warming in the Antarctic Peninsula (Turner et al., 2016; Thomas et al.,
2009) and West Antarctica during the 20th century. Contemporaneously, Antarctic sea ice is also undergoing
significant change. Despite model predictions of a homogeneous decline (Roach et al., 2020), total Antarctic sea
ice cover has increased since 1970 (Zwally et al., 2002; Turner et al., 2009). With more recent periods of abrupt
decline in 2016, (Meehl et al., 2016) and 2022 (Turner et al., 2022).
Our understanding of winds, atmospheric circulation and sea ice is hampered by both the lack of observations
prior to the instrumental period (~1970s onwards) and uneven spatial coverage of paleoclimate records (Jones et
al., 2016; Thomas et al., 2019). Data-model intercomparison and data synthesis studies have demonstrated the
value of large datasets in reconstructing climate and sea ice over decadal to centennial scales (Dalaiden et al.,
2021; Fogt et al., 2022). To meet the need for Antarctic-wide, spatially dense, and intercomparable atmospheric
circulation and sea ice records, we propose the use of chemical species routinely measured in ice cores.
Sodium [Na$^+$], from sea salt aerosol, has been proposed as a proxy for past sea ice extent (SIE)
(Waisdivideprojectmembers et al., 2013; Severi et al., 2017; Wolff et al., 2006; Winski et al., 2021). The sea
salt component of [Na$^+$] arises from both sea ice and open water and the relationship between [Na$^+$] and sea ice
varies between sites (Sneed et al., 2011). High winds mobilize [Na$^+$] from the sea ice surface, either in frost
flowers or brine-soaked snow (Huang and Jaeglé, 2017; Frey et al., 2020). The [Na$^+$] reaching the ice core sites
is dependent on both the distances from the source, either sea ice or open ocean, and the meteorological
conditions (Minikin et al., 1994). [Na$^+$] is therefore a valuable tracer for marine-air mass advection and changes
in atmospheric circulation (Dixon et al., 2004; Mayewski et al., 2017).
Sulphate [SO$_4^{2-}$] is formed in the atmosphere as secondary aerosol following volcanic and anthropogenic
sulphur dioxide [SO$_2$] gas emissions. [SO$_4^{2-}$], together with methane sulphonic acid [MSA$^-$], is the main
atmospheric sulphur compound formed from ocean-derived dimethylsulfide (DMS) (Gondwe et al., 2003). In
the southern hemisphere, marine biogenic emissions dominate the total sulphur budget (Delmas et al., 1982;
Legrand and Mayewski, 1997; Mccoy et al., 2015). Sulphur can significantly impact cloud albedo and new
particle formation (Brean et al., 2021). The sea salt fraction of [SO$_4^{2-}$] is largest at coastal and low elevation sites
(Dixon et al., 2004). The non-sea salt fraction, also referred to as excess [SO$_4^{2-}$] (hereafter referred to as xs
[SO$_4^{2-}$]), can be estimated based on the relationship with [Na$^+$] (e.g., xs [SO$_4^{2-}$] = [SO$_4^{2-}$] −0.25[Na$^+$])(O'brien et
al., 1995). Excess [SO$_4^{2-}$] has been shown to correlate with SIE at some ice core sites (Dixon et al., 2004; Sneed
et al., 2011). The background xs [SO$_4^{2-}$] source, from marine biogenic deposition, is superimposed by sporadic
volcanic deposition of [SO$_4^{2-}$] providing an excellent reference horizon for dating Antarctic ice cores (Dixon et
al., 2004; Sigl et al., 2014; Plummer et al., 2012). At low elevation and coastal sites, where background biogenic
sources are high, it is not always possible to identify volcanic peaks (Emanuelsson et al., 2022; Tetzner et al.,
2021b). In this study, [SO$_4^{2-}$] provides a dual function: 1) as a potential proxy for SIE and 2) as a stratigraphic
age marker to validate submitted age-scales and subsequently align ice-core chronologies onto a common
chronology.

### 1.1. The CLIVASH2k chemistry database.



CLIVASH2k (CLimate Variability in Antarctica and the Southern Hemisphere over the past 2000 years) is a
project of the Past Global Changes (PAGES) 2k network. The CLIVASH2k database is the latest in a series of
community-led paleoclimate data synthesis efforts endorsed by PAGES (Kaufman et al., 2020; Mcgregor et al.,
2015; Mckay and Kaufman, 2014; Tierney et al., 2015; Thomas et al., 2017; Stenni et al., 2017; Konecky et al.,
2020).  The aim of this study is to focus on two primary species, sodium, and sulphate, as they are routinely
measured in ice cores and have potential links with either sea ice or atmospheric circulation. The time window
of the last 2000 years has been selected to cover both natural and anthropogenic changes.
Two main features distinguish the CLIVASH2k data compilation from previous PAGES synthesis: 1) the data
included are not limited to previously published records, and 2) the data comprise two distinct chemical species
which do not have a well-established relationship with climate (beyond the episodic sources of [$SO_4^{2-}$] noted
above).
Calls for participation in CLIVASH2k activities were widely distributed, ensuring a representative cross section
of scientists from various disciplines, geographic regions, and career stage. The targeted species to target and the
selection criteria were decided at several open discussion stages, followed by updates to the CLIVASH2k
mailing list and distributed via PAGES monthly updates.
**2.   Methods**
**2.1. Resolution and duration.**
The target time-period for the database is the last 2000 years. Records of any duration could be submitted within
this time-period. These records could be from snow-pits and firn cores, spanning just a few seasons to years.
Data were requested at the highest resolution available and converted to annual averages (January – December).
Years with missing data were included, providing a threshold of half a year of data was achieved.
**2.2. Age-scales.**
Most records within this time-period have been annually dated, based on the seasonal deposition of distinct
chemical species (including sodium, sulphate, and sulphur). The longer records, those spanning the last 500-
2000 years, have been synchronized previously (Sigl et al., 2014) or within this project on the WD2014 age-
scale (Sigl et al., 2016) or have age-scales that are broadly consistent with WD2014 (Plummer et al., 2012). This
new chronology is constrained by the 774 CE cosmogenic (i.e. [10]Be) anomaly, and is consistent with
dendrochronology (Büntgen et al., 2018) and ice core chronologies from Greenland (Sigl et al., 2015). The
WD2014 age-scale is recommended because it is consistent with the forcings applied in PMIP4/CMIP6 model
simulations (Jungclaus et al., 2017). Age transfer functions can now be linked to other PAGES2k
reconstructions and individual ice cores. There are two exceptions, Plateau Remote and DT401 (both very low
accumulation sites in the interior of east Antarctica), which differ from WD2014 prior to 1000 AD and cannot
be confidently synchronized. The third exception is partly unpublished data from the Vostok vicinity, which
were dated using the snow accumulation rate and volcanic age markers (this study and (Ekaykin et al., 2014).
**2.3. Peer review and publications.**
Unlike previous PAGES 2k compilations, the CLIVASH2k database was not constrained by the need for
records to be published and peer reviewed. This decision arose based on the limited number of published
chemistry records available and the desire to maximise the records. Published records were submitted along
with their original citation; unpublished records were listed as "This study", with the data contributor included
as a co-author.
**2.4. Analytical methods.**
Both the ionic and elemental forms of sodium ([Na] and [$Na^+$]) and sulphur ([S] and [$SO_4^{2-}$]), respectively, were
accepted as part of the CLIVASH2k data call. Several analytical techniques are used to measure [$Na^+$], [S] and
[$SO_4^{2-}$] in ice cores. Ionic [$Na^+$] and [$SO_4^{2-}$] are typically measured by ion chromatography (IC), while elemental
Na and S are generally measured by inductively coupled plasma mass spectrometry (ICP-MS). Unlike IC, which
measures the soluble fraction, ICP-MS techniques measure the total elemental concentration of both the
dissolved and particulate fraction of the element. While continuous ICP-MS measurements of certain species



may require correction for under-recovery, Na and S are typically fully recovered during continuous
measurements (Arienzo et al., 2019). Previous comparisons of analytical methods show excellent agreement of
[Na] in ice cores measured using IC and ICP-MS methods e.g. (Grieman et al., 2022). This agreement suggests
that the ionic and elemental forms reported in the database can be directly compared.
Biogenic atmospheric emissions of organic [S] species, mainly dimethyl sulfide (DMS), are a major contributor
to the [S] in the Antarctic snow. In the marine atmosphere DMS is oxidized to [$MSA^-$] and [$SO_4^{2-}$], which are
eventually deposited on the polar ice sheets. The ICP-MS technique measures total [S] in ice cores, which
includes [S] contained [$MSA^-$]. In contrast, the IC technique solely quantifies [S]. If total [S] and [$MSA^-$] are
both analysed on the same ice core, the [$MSA^-$] contribution can be subtracted(Cole-Dai et al., 2021). However,
continuous [$MSA^-$] measurements are scarce over Antarctica (Thomas et al., 2019) and the long-term variability
of both [$MSA^-$] and [$SO_4^{2-}$] is very small during the common era (Legrand et al., 1992; Saltzman et al., 2006).
Thus, we applied a consistent transformation across all sites. We multiplied elemental [S] (32 g $mol^{-1}$) from
ICP-MS measurements with three to convert to the equivalent [$SO_4^{2-}$] (96 g $mol^{-1}$) without applying corrections
for MSA contributions. To aid ease of comparison, all [S] has been converted to [$SO_4^{2-}$], in the database and will
be referred to only as [$SO_4^{2-}$] in the data description.
**2.5. Flux vs concentration.**
[$Na^+$] and [$SO_4^{2-}$] in ice cores are generally reported as a concentration. Concentration can be converted to a
deposition flux, provided that the snow accumulation rate is known. The CLIVASH2k database includes both
concentrations and fluxes, when available. Flux estimates from ice cores combine both wet and dry deposition,
of which the contribution of these two depositional modes varies across Antarctica with elevation and distance
from the source (Wolff, 2012).
**2.6. Establishing the sea salt and non-sea salt component.**
There are various methods of calculating the sea salt (ss) and excess (xs) components of an ice core chemistry
record. The most-common method, as mentioned above, is to assume 100% of the [$Na^+$] comes from the ocean.
Then [$Na^+$] can be treated as a marine reference species and the ss fraction of all other chemical species can be
calculated based upon a mean ocean water elemental abundance reference value (e.g. (Lide, 2005). If [$Na^+$] is
suspected of not being of marine origin, alternative methods of calculating the ss chemical fraction may be
employed. For example, one may apply a standard sea-water ratio of 30.61 [$Na^+$], 1.1 [$K^+$], 3.69 [$Mg^{2+}$], 1.16
[$Ca^{2+}$], 55.04 [$Cl^-$] and 7.68 [$SO_4^{2-}$] to the ion concentrations in each sample (Holland, 1978). Several studies
have shown that frost flowers are depleted in [$SO_4^{2-}$] relative to [$Na^+$]. This produces a $ssSO_4^{2-}$ value which is
slightly higher than it should be for sites near the coast (Rankin et al., 2002; Rankin et al., 2000). Unfortunately,
not all studies accurately measure a wide suite of chemical species. Therefore, in this study we have assumed
[$Na^+$] to be the primary marine species and calculated xs [$SO_4^{2-}$] according to the following ratio: [$xsSO_4^{2-}$] =
[$SO_4^{2-}$] $-0.25$[$Na^+$](O'brien et al., 1995). Other ratios may be more suitable for coastal sites, but for consistency
we have applied the same ratio to all records reported in the database.
**2.7. Data validation and recommendations**
The two main uncertainties in the data presented arise from 1) chronological controls and 2) analytical errors.
As discussed in section 2.2, all records have been synchronised to a common age-scale (WD2014). Thus, when
using the entire database, we recommend using an error estimate of ±2 years, for records younger than 500
years, increasing to a conservative error estimate of ±5 years for records extending to 2000 years. This is the
maximum uncertainty estimate for the WD2014 age-scale at 2,500 years (Sigl et al., 2015). However, we note
that for individual records in this database the published error estimates are as low as ±1 year (e.g., Emanuelsson
et al., 2022). When using individual records we recommend using the published error estimate for that record.
Analytical precision varies between instruments and laboratories. We recommend applying a 1 standard error
($\sigma$) to the data to account for analytical errors.
The [$Na^+$] and [$SO_4^{2-}$] data is an accurate representation of either concentration or flux at a certain site.
However, how this relates to regional deposition is not well constrained. While we can account for the
uncertainty in analytical precision and dating error, we cannot define the signal to noise ratio associated with





small scale post-depositional process. For example, wind redistribution or the impact of local orography. The
regional climate and signal to local noise has been investigated for stable water isotopes in Antarctica (e.g.,
Munch et al., 2018), however, a detailed investigation of [$Na^+$] and [$SO_4^{2-}$] is lacking. One of the main
limitations, which this database will address, has been the lack of available data. We thus encourage database
users to investigate the regional signal by averaging records to reduce the signal to noise ratio. In this case, we
recommend using the standard error propagation procedure for averaging for example the square root of the sum
of variances of individual records divided by the number of the records.
Ice cores provide the only record of [$Na^+$] and [$SO_4^{2-}$] deposition in Antarctica, and therefore, validation against
reference datasets is also not possible. While progress has been made using chemical transport models to
represent the deposition of sea salts in Greenland (Rhodes et al., 2018), the period examined is very short
(annual to decadal) and has currently not been applied to Antarctica. This database will provide much needed
data for any future model validation. However, currently it means there are no independent data products to
validate our [$Na^+$] and [$SO_4^{2-}$] records against.

## 3. Data records

A total of 117 records were submitted, representing 105 individual ice core sites (Fig. 1). In some locations,
duplicate analysis or updated versions were submitted (e.g. EPICA Dome C). All submitted records have been
included in the database. The number of records submitted is summarised in Table 1. The full list of records,
their location, elevation, duration, and reference are presented in appendix A (table A1).
**Table 1.** Summary of records submitted to the CLIVASH2k database.

| | [$Na^+$] | | [$SO_4^{2-}$] | | xs [$SO_4^{2-}$] | |
|---|---|---|---|---|---|---|
| | **Concentration** | **Flux** | **Concentration** | **Flux** | **Concentration** | **Flux** |
| **Total records** | 105 | 68 | 103 | 67 | 95 | 61 |
| **Duplicates** | 15 | 4 | 13 | 2 | 11 | 2 |
| **Total ice core sites** | 90 | 64 | 90 | 65 | 84 | 59 |

### 3.1. Geographical and temporal coverage.

There is reasonable spatial coverage across Antarctica, with the largest density of records in West Antarctica
(Figs. 1a & 1b). In East Antarctica, notable data voids include Coats Land, Enderby and Kemp Land, Wilkes
Land and Terra Adelie. There is a notable absence of long records from the Antarctic Peninsula. Despite the
high density of records in West Antarctica, high snow accumulation in this region results in most of these
records only spanning the last few decades or centuries.
The longer duration records (>1000 years) are predominantly found in the central East Antarctic plateau, while
most higher snow accumulation coastal sites cover shorter timescales (Figs. 1a &1b). The most recent year in
the record peaks in the late 1990s, when the highest number of cores were drilled (Figs, 1c & 1d). Only eleven
records span the full 2000 years.

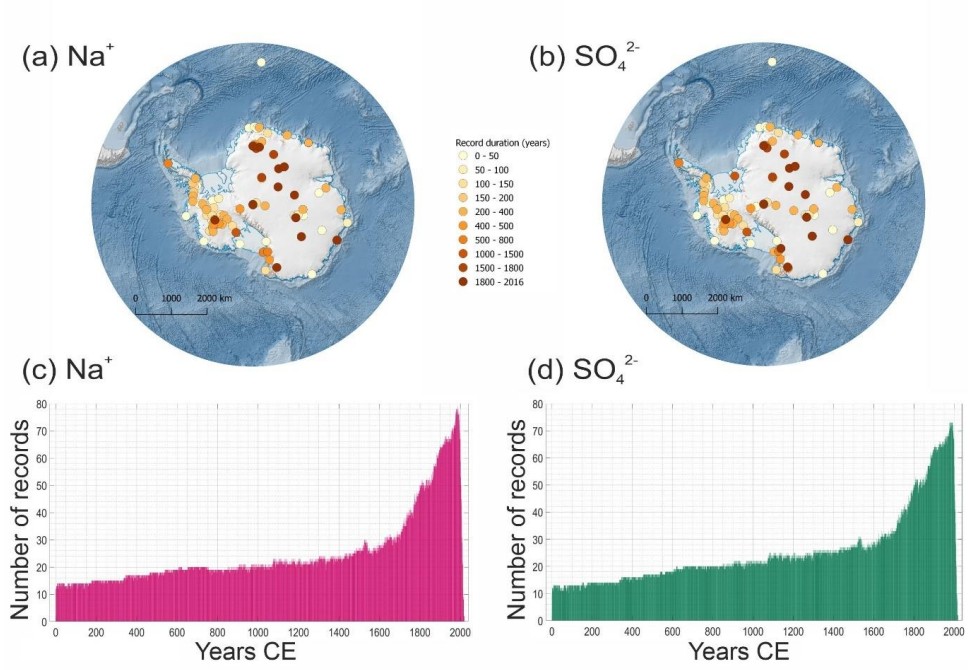

**Figure 1.** Spatial and temporal coverage of records in the CLIVASH2k database. Map of ice core locations with (a) [$Na^+$], and (b) [$SO_4^{2-}$] records. Colour coded based on record duration (number of years). The number of (c) [$Na^+$] and (d) [$SO_4^{2-}$] records as a function of the years (CE) covered.

### 3.1.1.    Technical validation

To facilitate the scientific usability of this database, we have evaluated each record in terms of its relationship with key climate parameters during the observational period (1979- 2019). Given their varying temporal ranges (Fig. 1), not all the records span the full satellite period. Thus, correlations are based on the largest number of years available within this period. Although the database includes short records, for the data interpretation step, we have only included records that have at least ten years of overlap with the satellite and reanalysis climate data. Duplicate records (including updated versions and different analytical approaches) are included in the data interpretation step and interpreted as individual records.

The objective of this climatological comparison is to provide a first level filter for the database. Thus enabling database users to quickly search for sites that exhibit a direct and dynamically logical relationship with sea ice concentration (SIC), wind conditions and atmospheric circulation to facilitate future investigations.

All of the records were also correlated using ERA5 meteorological parameters (Hersbach et al., 2020), the fifth generation European Centre for Medium Range Weather Forecast (ECMWF) atmospheric reanalysis data. These parameters include 500-hPa geopotential height (Z500), meridional winds (v) and zonal winds (u) both at the 850-hPa level. The 850 hPa level was chosen to represent surface winds (relevant for sea ice reconstructions), while the 500 hPa was chosen to capture larger-scale circulation across both high and low elevation sites. All correlations were performed on de-trended annual average data (January – December) to correspond with the annually-resolved ice core records and corrected for autocorrelation. All of the records were correlated with SIC from the National Snow and Ice Data Centre (NSIDC) Nimbus-7 SMMR and DMSP SSM/I-SSMIS Passive Microwave Data version 1 (Cavalieri et al., 1997).

## 4. Data interpretation
### 4.1. Identifying sites that correlate with sea ice and atmospheric circulation

An example of the data interpretation output is presented in Figure 2. For consistency, correlations were performed with climate variables across all longitudes in the southern hemisphere south of 50ºS. This approach has the potential to generate spurious results or correlations in regions that are physically unrelated to the site (e.g., Fig. 2b). Therefore, each record was individually evaluated by an expert (hereafter the "interpretation team") to establish if the correlations observed can be attributed to a realistic source region and transport mechanism. Sites with a clear connection or absence of connection agreed by more than one interpreter were marked as either "yes" or "no" (Figs. 2a & 2b). Sites where the transport mechanism was less clear, or there was a disagreement between interpreters were listed as "uncertain" (Fig. 2c).

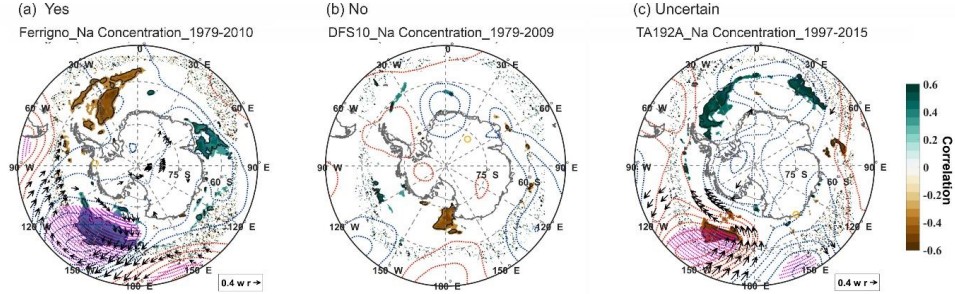

**Figure 2.** Example correlation plots evaluated by the data interpretation team. (a) Yes example, correlation observed between all three parameters. (b) No example, no significant correlation observed with any parameters. In this example, a significant correlation with SIC at a distant location is likely an auto-correlation artefact. (c) Uncertain example, the transport mechanism could not be verified by the interpretation team. Yellow open circle indicates ice core location. Coloured shading indicates positive (green) and negative (brown) correlations with SIC (data from NSIDC), solid black line correlations significant at the 5% level. Correlations with winds (arrows) composed of u850 and v850 (ERA 5). Dashed red and blue contours represent positive (red) and negative (blue) correlations with geopotential height at 500 hPa (ERA5), pink hatching is significant at the 5% level. Plot titles labelled as "Site name_species_years for correlation".

In the following sections, we only refer to records that exhibited a correlation that is statistically significant at the 5% level ($p<0.05$) (hereafter referred to as significant). For sites to be identified as having a relationship with either SIC, atmospheric pressure (z500) or winds (u850 or v850), they had to be supported by a valid transport mechanism or source region as evaluated by the data interpretation team (Fig. 2). We have not applied a uniform cut-off size for the area of correlation or specified a minimum or maximum distance from the source region, as these features will be site specific. For example, a low elevation coastal site (e.g., Sherman Island) may only capture local changes in sea ice that will appear as a small area of correlation on the map (e.g., (Tetzner et al., 2021a) while a central Antarctic site (e.g., South Pole) might be influenced by long-range air-masses and thus capture changes in sea ice from a relatively distant source region e.g., (Winski et al., 2021).

The database contains more concentration records than flux records. Thus, in the data interpretation we presented both the total number of sites, and the proportion of sites, that exhibit a significant correlation with meteorological parameters. The total number of eligible records for each species is shown in Table 3. The spatial distribution of records is presented in figures 3, 4 and 5.

**Table 3.** Summary of the number of records that display a significant correlation (5% level) with SIC, wind fields (meridional (v850) and zonal (u850)), and geopotential height (z500). The total records available for the data interpretation step is shown for each species. This includes all records with more than 10-years overlap



with the instrumental period (1979-2018) and includes duplicates. Brackets indicate the number of sites marked
as "uncertain". The percentage of records shown in italics underneath to account for the varying sample size.

| Variable | [Na$^+$] | Na$^+$ Flux | [SO$_4^{2-}$] | SO$_4^{2-}$ Flux | xs [SO$_4^{2-}$] | xs SO$_4^{2-}$ Flux |
|---|---|---|---|---|---|---|
| Total records | 88 | 65 | 84 | 61 | 81 | 59 |
| SIC | 69 (6) *78 %* | 56 (4) *86 %* | 60 (6) *71 %* | 40 (5) *66 %* | 68 (5) *84 %* | 42 (2) *71 %* |
| Wind (v850 or u850) | 63 (3) *72 %* | 48 (4) *74 %* | 54 (8) *64 %* | 39 (3) *64 %* | 56 (3) *69 %* | 40 (3) *68 %* |
| Geopotential Height (z500) | 47 (2) *53 %* | 43 (3) *66 %* | 38 (6) *45 %* | 26 (3) *43 %* | 40 (6) *49 %* | 23 (3) *39 %* |

**4.2. Sodium (concentration and flux)**
A total of 69 (out of 88) [Na$^+$] sites exhibit a correlation with SIC, with an additional six records marked as
"uncertain" (Table 3). Fifty-Six (out of 65) records are correlated with SIC when using Na$^+$ flux, with an
additional four sites marked as uncertain. This reflects the smaller number of flux records submitted to the
database. Proportionally, more records are correlated with SIC when using flux that concentration (78 %
compared to 72 %).
A total of 63 (out of 88) [Na$^+$] records exhibit a significant correlation with the wind fields (v850 and u850).
While an additional four records were marked as uncertain. When using Na$^+$ flux 48 (out of 65) records
correlated with winds, with four records marked as uncertain. A higher proportion of records (74 % compared
with 72 %) correlated with winds when using flux.
A total of 47 (out of 88) [Na$^+$] sites exhibit a significant correlation with geopotential height. While an
additional two records are marked as uncertain. The number of correlations with geopotential height is 43 (out
of 65) when using Na$^+$ flux, with an additional three sites marked as uncertain. A higher proportion of records
(66 % compared with 53 %) correlated with atmospheric circulation when using flux.

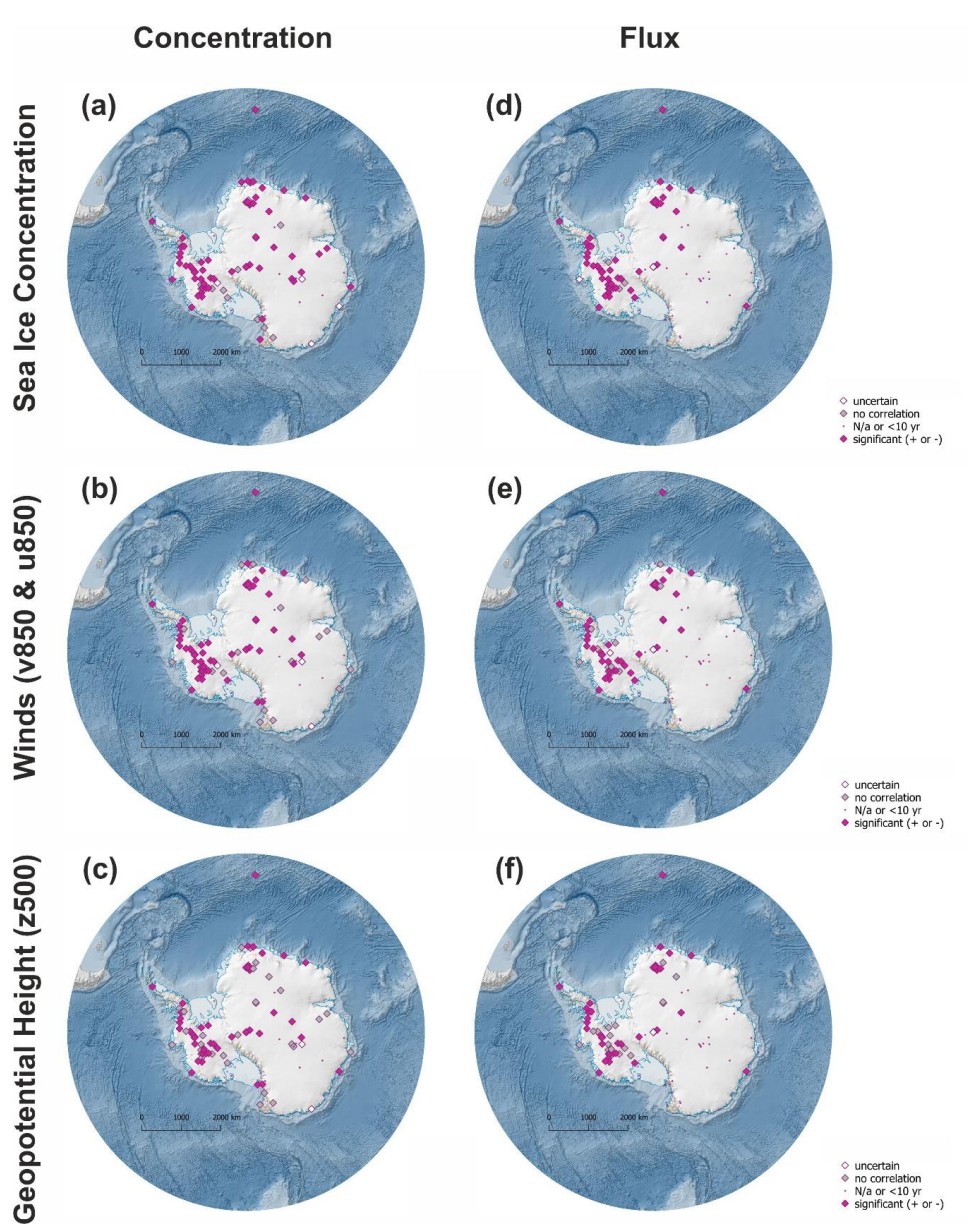

**Figure 3** – Geographical distribution of [Na$^+$] records (left column) which exhibit a statistically significant (p>0.05) correlation with (a) SIC, (b) winds (v850 and u850) and (c) geopotential height (z500). Compared with the geographical distribution of Na flux record (right column) which exhibit a statistically significant (p>0.05) correlation with (d) SIC, (e) winds (v850 and u850) and (f) geopotential height (z500). Pink diamonds are locations with a significant correlation either positive or negative; grey diamonds are sites with no correlation, open diamonds are uncertain. Dots indicate ice core locations that are in the database but either are less than 10 years in length (or overlap with the instrumental period) or sites which failed to generate any correlations with parameters tested.





### 4.3. Sulphate (concentration and flux)

A total of 60 (out of 84) $[SO_4^{2-}]$ records display a correlation with SIC, with six additional records marked as uncertain (Table 3). When using $SO_4^{2-}$ flux, 39 (out of 61) records correlated with SIC, with an additional five records marked as uncertain. A slightly higher proportion of records (71 % compared with 64 %) correlated with SIC when using flux.

Fifty-four $[SO_4^{2-}]$ records (out of 84) are correlated with winds (v850 and u850), with eight additional records marked as uncertain. This is compared to 39 records (out of 61), and three additional records marked as uncertain, that are correlated with winds when using $SO_4^{2-}$ flux. The proportion of records correlated with winds (64 %) is the same when using either flux or concentration.

A total of 38 (out of 84) $[SO_4^{2-}]$ records are correlated with geopotential height, with six additional records marked as uncertain. This is compared with 26 records (out of 59) when using flux, with three marked as uncertain. A slightly higher proportion of records (45 % compared with 43 %) are correlated with atmospheric circulation when using flux.

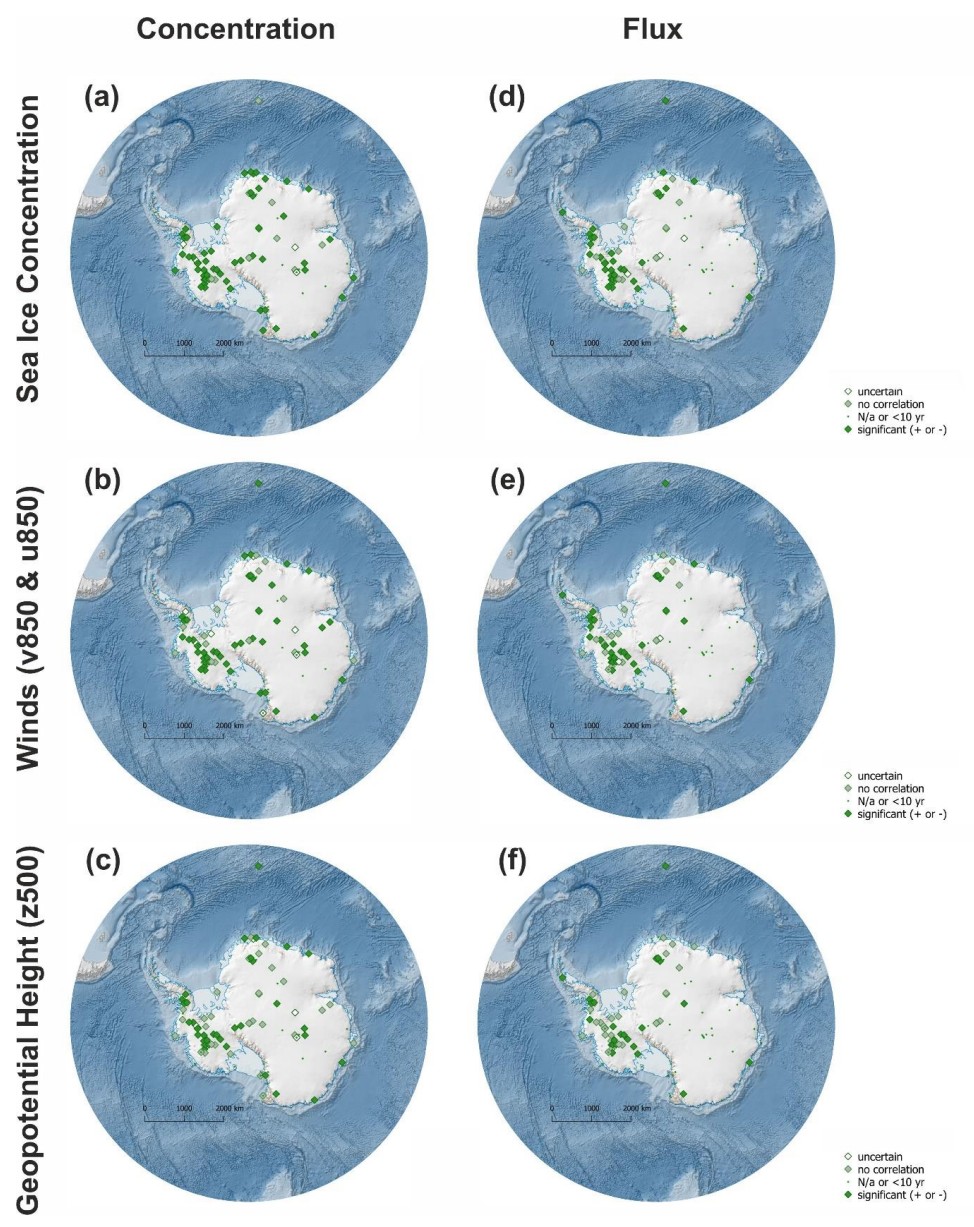

**Figure 4** – Geographical distribution of [$SO_4^{2-}$] records (left column) which exhibit a statistically significant (p>0.05) correlation with (a) SIC, (b) winds (v850 and u850) and (c) geopotential height (z500). Compared with the geographical distribution of $SO_4^{2-}$ flux record (right column) which exhibit a statistically significant (p>0.05) correlation with (d) SIC, (e) winds (v850 and u850) and (f) geopotential height (z500). Green diamonds are locations with a significant correlation either positive or negative; grey diamonds are sites with no correlation, open diamonds are uncertain. Dots indicate ice core locations that are in the database but either are less than 10 years in length (or overlap with the instrumental period) or sites which failed to generate any correlations with parameters tested.





### 4.4. Excess Sulphate (concentration and flux)

A total of 68 (out of 81) xs [$SO_4^{2-}$] records are correlated with SIC, with five additional records marked as uncertain when using concentration (Table 3). This number drops to 42 (out of 59) when using the flux, with two additional records marked as uncertain. A smaller proportion of records (71 % compared with 84 %) correlated with SIC when using flux.

A total of 56 (out of 81) xs [$SO_4^{2-}$] records are correlated winds (v850 and u850), with three additional records marked as uncertain. The number drops to 40 (out of 59) records when using the xs $SO_4^{2-}$ flux, with three additional records marked as uncertain. A smaller proportion of records (68% compared with 69 %) correlated with winds when using flux.

A total of 40 (out of 81) xs [$SO_4^{2-}$] concentration records are correlated with geopotential height, with an additional six records marked as uncertain. The number drops to 23 (out of 59) records when using the xs $SO_4^{2-}$ flux, with three additional records marked as uncertain. A smaller proportion of records (39 % compared with 49 %) correlated with atmospheric circulation when using flux.

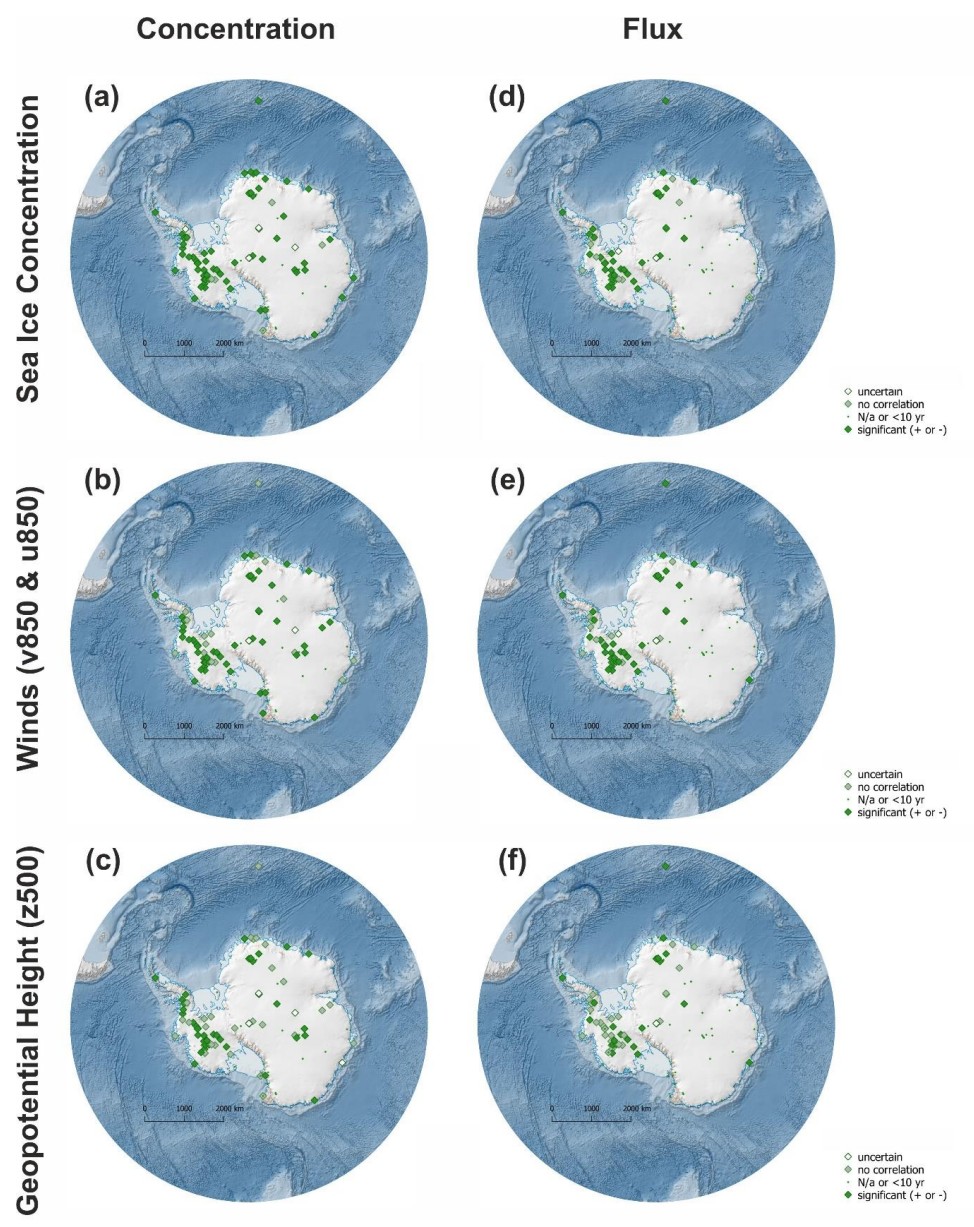

395

**Figure 5** – Geographical distribution of xs [$SO_4^{2-}$] records (left column) which exhibit a statistically significant (p>0.05) correlation with (a) SIC, (b) winds (v850 and u850) and (c) geopotential height (z500). Compared with the geographical distribution of xs $SO_4^{2-}$ flux record (right column) which exhibit a statistically significant (p>0.05) correlation with (d) SIC, (e) winds (v850 and u850) and (f) geopotential height (z500). Green diamonds are locations with a significant correlation either positive or negative; grey diamonds are sites with no correlation, open diamonds are uncertain. Dots indicate ice core locations that are in the database but either are less than 10 years in length (or overlap with the instrumental period) or sites which failed to generate any correlations with parameters tested.

404





## 5. Discussion

### 5.1. Which records are suitable for reconstructing SIC, winds and atmospheric circulation?

Our findings reveal that [Na] provides the highest number (69) of records that exhibit a significant correlation with SIC. Only fractionally higher than the number of xs [$SO_4^{2-}$] records (68) and SO4 (60). Thus, all three records have the potential to capture changes in sea ice conditions. The full list of which sites exhibit positive correlations with each parameter is shown in Supplementary Figure S2.

[Na] also provides the highest number of correlations with geopotential height (47) and wind (63). However, proportionally Na flux has the highest number of correlations with geopotential height and winds. While less than 49% of the [$SO_4^{2-}$] and xs [$SO_4^{2-}$] data exhibit relationships with geopotential height, a much higher percentage (64-69 %) display correlations with winds. This suggests that there is greater potential for using [$SO_4^{2-}$] and xs [$SO_4^{2-}$] for reconstructing winds and SIC than geopotential heights. Removing the sea-salt component of [$SO_4^{2-}$] to produce xs [$SO_4^{2-}$] improves the relationship with SIC, geopotential height and winds.

Most of the records from West Antarctica and the Antarctic Peninsula (both [$Na^+$] and [$SO_4^{2-}$]) exhibit correlations with SIC, geopotential height and winds. This reflects the dominance of marine air-mass incursions in this region (Suzuki et al., 2013), transporting sea salt aerosols from the sea ice zone to the ice core sites. In East Antarctica, the high elevation of the ice sheet (>3000 m) acts as a barrier to marine air-mass transport. However, this study corroborates previous studies (e.g., (Winski et al., 2021)) suggesting that [$Na^+$] and [$SO_4^{2-}$] concentrations from ice cores in the East Antarctic plateau are significantly correlated with SIC and atmospheric circulation.

Converting the records to flux drastically reduces the geographical coverage. In most cases this is due to the lack of available snow accumulation records from central Antarctica to covert to flux. However, our study demonstrates that converting [$Na^+$] to flux increases the relative proportion of records that exhibit a significant correlation with SIC, geopotential height and winds. The opposite is true for [$SO_4^{2-}$] and xs [$SO_4^{2-}$], which results in a lower proportion of records correlating with SIC after converting to flux. This may suggest a dominance of wet deposition of [$Na^+$] and dry deposition of [$SO_4^{2-}$]. However, a detailed evaluation of the relationships between ion concentration and snow accumulation is needed to address this fully.

Overall, [Na] provides the most records which exhibit significant correlations across all three parameters (179), followed by xs [$SO_4^{2-}$] (164) and [$SO_4^{2-}$] (152).

### 5.2. Potential limitations

There are limitations to this assessment, which is intended as a first pass filter to highlight the potential future use of the data. In particular, the numbers only relate to records that span or have at least 10-years of data that overlap with the instrumental period. This is defined as the period from 1979-2019 and accounts for 88% of the records (438 out of 499 records submitted). Thus, relationships may exist for shorter records or records drilled prior to 1979, however, it is not possible to verify this under the defined criteria. Another caveat is that correlations have only been conducted with a single sea ice (NSIDC) and reanalysis (ERA-5) product, and results may vary with different datasets. Results may also be impacted by the different timespans used. For example, it was not possible to select the same reference period to run all correlations, because record lengths and top ages (date the core was drilled) vary considerably. Thus, the assumed stationarity in the source and transport routes may not be appropriate.

We also note that almost 8% of the records have been classified as "uncertain". In some cases, significant correlations were evident in the plots, but they were difficult to explain (Fig. 2c). For example, Law Dome generates several regions of significant correlations across multiple sectors, however not in the ocean adjacent to the site. This may indicate long-term transport or the influence of large-scale atmospheric circulation (e.g., SAM, Indian Ocean Dipole, Atlantic Multidecadal Oscillation). However, in this first pass filter we only included sites where a clear mechanism was evident.

## 6. Data availability



This data descriptor presents version 1.0.0 of the CLIVASH2k Antarctic ice core chemistry database PAGES
CLIVASH2k database (Thomas et al., 2022). The database can be accessed via the UK Polar Data Centre.
NERC EDS UK Polar Data Centre. https://doi.org/10.5285/9E0ED16E-F2AB-4372-8DF3-FDE7E388C9A7
**7.  Conclusions.**
The CLIVASH2k database is the first attempt to compile an Antarctic continental-scale database of chemical
records in ice cores spanning the past 2000 years. This study is the first phase of the project, the goal of which
was to compile and publish the records. In this study we have provided all available $[Na^+]$ and $[SO_4^{2-}]$ records
submitted by the community. The records are all available as annual averages, included as both concentration
and flux (if available). An additional parameter, xs $[SO_4^{2-}]$ has also been calculated where possible.
To facilitate future data interpretation, we have run spatial correlations for all the records. The aim of this
analysis is to identify sites which exhibit a statistically significant relationship with sea ice concentration (SIC)
and atmospheric circulation (500-hPa geopotential heights) or winds (v850 and u850). This is intended as a first
filter to identify potential records that could be used in future proxy reconstructions.
This first pass filter demonstrates that when considering the species separately, 335 individual records exhibit
statistically significant correlations with SIC that have been verified by a team of experts. A recent compilation
of available ice core derived sea ice reconstructions, based on a range of proxy data, identified only 17
individual sites which have been used to reconstruct sea ice (Thomas et al., 2019). Thus, this data compilation
represents a significant improvement on existing published or available data.
For researchers interested in reconstructing winds or atmospheric circulation the CLIVASH2k database contains
a total of 300 records that are significantly correlated with the wind fields (v850 and u850) and 217 records that
are significantly correlated with geopotential height (500 hPa). The Na flux exhibits the greatest proportion of
records that correlate with sea ice, atmospheric circulation, and winds. This analysis suggests that $Na^+$ flux may
be the best proxy for reconstructing all three parameters.
Future work will focus on using this database to:
1)  Investigate the deposition of $[Na^+]$ and $[SO_4^{2-}]$ over decadal to centennial timescales.
2)  Provide a reconstruction of sea ice or atmospheric circulation spanning the past 2000 years.
3)  Evaluate the skill of chemical transport models to capture observed deposition of $[Na^+]$ and $[SO_4^{2-}]$.
4)  Combine the information in this new database with the database of snow accumulation (Thomas et al.,
2017) and isotopic content (Stenni et al., 2017) to obtain a comprehensive view of Antarctic climate
variations over the past 2000 years.
This is not an exhaustive list, and we encourage the community to engage with the CLIVASH2k working group
and make use of the database.
**Author contributions**
ET and HG conceived the idea. ET & DV initiated the data call and coordinated the project. ET wrote the paper
with contributions from the core writing group. The core writing group (DV, ACFK, DE, HG, DW, VHLW,
DD, DU, TV), contributed to the paper writing and discussions. The data interpretation team (ET, DV, ACFK,
DW, VHLW, DD, NC, DU, TV, DT, MMG, MS) quality checked the data, evaluated the age-scales, and
interpreted the spatial correlation plots. NANB, AH, CML, JRM, YM, KT, HM, YN, FS, JCS, MS, RT, SW,
CX, JY, TVK, AAE, LPG and EMT all provided unpublished data. DE wrote the code for the data interpretation
plots. DV & LT compiled the figures. All authors read and commented on the manuscript.
The following researchers contributed published data to this database. Yoshiyuki Fujii, Lenneke Jong, Elisabeth
Isaksson, Filipe G. L. Lindau, Andrew Moy, Rachael Rhodes. We thank the many other researchers who have
already made their data available on public data repositories.
**Competing interests**
The authors declare no competing conflict of interest.



## Acknowledgements

CLIVASH2k is a contribution of Phase 3 of the PAGES 2k network. DT was funded as part of the PAGES Data
Stewardship scholarship awarded to ET. This financial support comes from the Chinese Academy of Sciences
(CAS) and the Swiss Academy of Sciences (SCNAT). We thank the PAGES office for their support and the
temporary data storage during the compilation of this database.

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
