# Peer review of "Ice core chemistry database: an Antarctic compilation of"

_Earth System Science Data, 2022_

## Author Comment (AC1)

**Response to reviewer 1:**

We thank the reviewer for their helpful comments and detailed suggestions. We are glad they have recognised the value and importance of this database. Below we respond to each point in turn.

General comment

**This paper represents the synthesis of a major community effort to create a useful and comprehensive dataset derived from hundreds of snowpits, and ice and firn cores retrieved over the past decades across Antarctica. Although some minor aspects listed below need to be clarified before publication, the article itself is adequate to support the publication of a dataset, and the overall structure of the article is well structured and clear to the reader. The annual resolution data are likely to be useful for future studies, thus I support their publication. I particularly appreciated the so-called "first-pass filtering" of the data provided, which makes it easier to identify the potential of a given ice core site for past environmental reconstructions and/or comparisons between data and models.**

Thank you for your comments. It was indeed our intention to apply a first-pass filter that would make it easier for the community to decide which records to use for future reconstructions.

**-Line 80: Please rephrase the sentence to avoid the repetition of the word "contain".**

Updated to "comprises".

**-Line 81: Specify which are the climate observation/variable used to filter the dataset.**

We have updated the sentence to specify which observations were used.

"An initial filter has been applied, based on evaluation against sea ice concentration, geopotential heights (500 hPa) and surface wind fields climate observations, to identify sites suitable for reconstructing past sea ice conditions, wind strength, or atmospheric circulation."

**-Line 96: Consider adding the words in capital for clarity:" value of large datasets in reconstructing climate and sea ice VARIABILITY over decadal to centennial TIME scales"**

Updated.

**-Line 100: Please fix the reference format for the "Wais divide project members".**

Updated.

**-Lines 105-106: Add additional Ref. to Minikin et al., 1994. A more recent study could be "Rhodes, R.H., Yang, X., Wolff, E.W., 2018. Sea ice versus storms: what controls sea salt in arctic ice cores? Geophys. Res. Lett. 45, 5572e5580."**

Included.

**-Line 114: In my opinion the annotation "xs" used to define the excess of [SO42-] is a bit confusing through the text. Please consider to change it with the annotation [SO42-]Exc according also to other ice-core based studies.**

We used the convention (xs) of Dixon et al., 2004 for consistency. This is the most comprehensive study of excess SO4 for our region of interest. We can change to excess throughout if preferred, however, we would rather stick with xs SO4.

**-Lines 132-134: The sentence "… 2) the data comprise two distinct chemical species which do not have a well-established relationship with climate (beyond the episodic sources of [SO42-] noted above)" is not clear, please rephrase it for clarity.**

The sentence has been expanded for clarity.

2) the data comprise two distinct chemical species which do not have a well-established relationship with climate. *This differs from previous compilations where the data can be either directly, or indirectly, compared with a modelled or observed climate parameter e.g. temperature (Stenni et al., 2017).*

**-Line 145: What do you mean for "… providing a threshold of half a year of data was achieved", please rephrase it for clarity.**

This sentence has been removed to avoid confusion. It only applies to the upper most (or lower most) years where the distance between the first age-marker and the surface (or bottom depth) is not a complete year.

**-Lines 156-158: Change "two" with "few" and "the third" with "another".**

Updated.

**-Lines 173-174. This statement is not entirely correct because is not enough to have an ICP-MS to see the total concentration. Once the sample is acidified, you can see part of the total, or the dissolved particles. In this regard, it is important to mention that there are different protocols for acidifying samples by making use of different acids (HCl, HNO3, HF, etc.) and different acidification times (days, weeks, months) based on which different absolute concentrations might be obtained as illustrated by Gaspari et al. (2006), Koffman et al. (2014), and Burgay et al. (2021).**

Thank you, this additional information is now included in this section.

"However, we note that there are different protocols for acidifying the samples prior to analysis which may result in different absolute concentrations, including the choice of acid, the acid concentration, and the acidification time.."

**-Lines 180-182: Insert Ref. to both sentences.**

Inserted Legrand and Mayewski, 1997 and Barnes et al., 2006.

**-Line 196: Insert the equation used to calculate flux from concentration with the units presented in the dataset (i.e. ppb kg m-2). It would also be appropriate to explain why these units were chosen instead of the more common (mg m-2) or (ug m-2). In this way, indeed, a user can easily infer the annual snow accumulation (in kg/m2 or mm w.e.) simply by dividing the flux with the concentration.**

Equation added.

If required, most of the snow accumulation records are available in the PAGES 2k database. We would strongly recommend using this database to extract snow accumulation, to ensure the correct citations are included.

Thomas, E. "Antarctic regional snow accumulation composites over the past 1000 years" v2 (2017) Polar Data Centre, Natural Environment Research Council, UK. doi:10.5285/cc1d42de-dfe6-40aa-a1a6-d45cb2fc8293

**-Lines 210: Add a Ref. to the sentence "Other ratios may be more suitable for coastal sites (Ref.), …".**

Added Dixon et al., 2004.

**-Lines 241-244: This part is a bit misleading. You start the paragraph saying, "A total of 117 records were submitted, representing 105 individual ice core sites (Fig. 1)", however, looking at Table 1, the Total records are only 105 and the Total ice core sites are 90 with 15 duplicates. Where are listed the remaining 12 records? Furthermore, please change "Table A1" with "Table S1" as indicated in Appendix A.**

We agree the table is a bit misleading and have updated it in the revised version.

There is a slight discrepancy in the definition of duplicates, which has now also been updated. There are 15 records which were submitted with the same location (same latitude and longitude). This includes 12 replicates, where the same ice core has been analysed using a different instrument, at a different resolution, at different laboratories or at a different time. However, at some sites (e.g., Vostok), there are multiple ice cores drilled at almost the exact same location (same coordinates) but are distinct and different ice cores. Hence why the total number of ice cores is 105.

We have separated out the different species in the new table, because not all the 117 records contained both Na and SO4. Some sites were submitted with just Na or just SO4, and not all replicates included both. We have also updated the table to clarify that a total of 94 ice core sites contain all three species (Na, SO4 and XSSO4) is 94.

Table reference changed and the analytical replicates are now highlighted in table S1 using italics.

We have also added the following line to the data records section.

"This includes sites where analysis was undertaken at different laboratories, using different instrumentation (e.g., IC and ICP-MS) or different depth resolution. Some ice cores only provide data for a single species and not all records contain both flux and concentration. A total of 94 ice core sites are included in the database which provide $[Na^+]$, $[SO_4^{2-}]$ and xs $[SO_4^{2-}]$."

**-Line 260 (Figure 1 / Panels A-B): Since all the records, except for one, are homogeneously distributed across all the continent it would be worth zooming in a little bit more and create a separate small box only for Bouvet Island.**

The figures have been updated to focus on the continent.

**(Figure 1 / Panels C-D): Because of the low resolution of the graphs, also after zooming in, it's still very hard to distinguish each single record based on its duration (the file looks like formatted as a jpg or jpeg file). Besides that, the right part of the graphs (i.e., around Year CE 2000) appears oblique going from the top to the bottom of the graph. Is it due to some indistinguishable records present from 2000 CE to the end of the graph? Please submit a new version of Panels C-D increasing the resolution (i.e., uploading the original vector file) for clarity.**

Figures all updated to improve resolution and clarity.

**-Lines 334-335: In the sentence "Proportionally, more records are correlated with SIC when using flux THAT (than?) concentration (78 % compared to 72 %)" percentages are not corrected. Please adjust them according to the values listed in Table 3 (if I understand correctly, it should be 86% compared to 78%).**

Corrected.

**-Line 338: The additional records marked as uncertain in Table 3 are 3 and not 4 as stated in the sentence. Fix the table or the sentence with the correct value.**

Corrected.

**-Line 349 (Figure 3): Similarly to what was suggested for Figure 1, in my opinion the panels shown in Fig. 3 would also deserve a zoom to better visualize the individual ice core sites and their spatial distribution across Antarctica (with a separate small box for Bouvet Island). Also consider changing the color of both the fill and the edges of the diamonds representing sites with no correlation; gray might be difficult to distinguish from sites with uncertain and/or significant correlation.**

All figures updated to better visualise the individual ice core sites.

**-Lines 361-362: The records of SO4 indicated in Table 3 are 40 and not 39 as stated in the sentence. Also, the percentage need to be fixed accordingly (they seem to be reversed in the sentence). Fix the table or the sentence with the correct value. Same in Line 369, "26 records (out of 59)" should be out of 61, as indicated in Table 3. Fix the table or the sentence with the correct value.**

Corrected.

**-Line 371: Flux should be concentration or, alternatively, the percentages must be inverted, and the sentence adjusted accordingly.**

Updated.

**-Lines 373-396 (Figure 4 and 5): Same comments as for Figure 3.**

Updated.

**-Line 407: For consistency, add the ionic form for [Na+] (also in lines 411,431, and 473).**

Updated.

**-Line 408: I would replace "thus" with "this suggest that".**

Updated.

**-Line 410: For an easier identification of the "most useful" sites for climatic studies, it would be nice if the full list of sites presented in Table S2 would be ordered from top to bottom according to the number of correlations with the others climate variables. In this way, sites with a larger number of correlations (i.e. "ITASE-02-6") would be listed in top rows of Table S2, while site with a lower number of correlations (i.e. WHG (ICPMS)) would fill the last rows.**

I agree that presenting the records in order of corelation makes a lot of sense, thank you. The table has been updated to list the sites with the most correlations first. An additional column is added to show the number of corelations for each site.

**-Lines 419-420: Replace "In East Antarctica" with "In contrast, in East Antarctica, "**

Updated .

**-Line 425: Replace "to covert to" with "needed to calculate the flux".**

Updated

**-Lines 431-432: Please rephrase this period for clarity. I suggest something like: "Overall, the records of [Na+] exhibit the highest number of correlations with the climatic variables considered (179 out of 264) , followed by xs [SO42-] (or [SO42-]Exc )(164 out of 243), and ([SO42-] (152 out of 252)."**

Updated.

**-Line 474: Rephrase the sentence about Na flux in a more conservative way, such as "The Na flux exhibits the greatest proportion of records that correlate with sea ice, atmospheric circulation, and winds. Therefore, among the ice-core chemical species considered in our analysis, we propose Na+ flux as the best candidate for reconstructing all three climatic components".**

Updated.

---

## Author Comment (AC2)

**Response to reviewer 2**

We thank the reviewer for their constructive comments and suggestions. We recognise that the interpretation is not fully explored in this publication. This is because the guidelines for a data description paper state that extensive interpretations of data remain outside the scope of this data journal. Therefore, we have focused on presenting the data to accompany the database and recognise the contributions of all co-authors who submitted previously unpublished data.

General comment

**Thomas et al present a unique data base of previously published and unpublished sodium [Na+] and sulphate [SO42-] records from all over Antarctica, collected within the CLIVASH2k project. The records span several 100 years back to 2000 years. Age scales were made consistent where possible and age transfer functions are provided. A data validation is provided and recommendations with respect to uncertainty. In a second step, a comparison to the climatological parameter sea ice condition, wind and geopotential height is conducted to serve as a first level filter for the data base. To this end the correlation of each record to the climatological parameter is evaluated.**

**The authors present the data set along with a careful description of uncertainty and potential usage. The provided data base presents a well needed contribution to data coverage related to Antarctic climate. The only weakness of the paper is the attempted interpretation, i.e. the presentation of the correlation analysis of single records.**

**Line 147ff: It would be beneficial to have the records published with both –age and depth. Having access to the original record over depths allows re-using the record and comparing it to new/upcoming records in the future.**

Most of the data does contain a depth scale, and it is our intention to include all files in LiPD format alongside the current database. However, not all the data (especially some of the historical records) contains a depth scale at this stage. Or not all authors were prepared to submit the records together. Therefore, we will work on this for future versions. However, while this step is ongoing, we felt it was important to make the data available to support existing community activities.

**Line 156: how are these cores dated/synchronized with the others?**

This sentence has been updated and a reference added for the DT401 age-scale.

"Plateau Remote and DT401, both very low accumulation sites in the interior of east Antarctica, have been dated using [SO42-] (Ren et al., 2010), however, the reference horizons differ from WD2014 age-scale prior to 1000 AD and cannot be confidently synchronized."

**In line 265ff (chapter 3.1.1) the correlation of single records with the three climatic parameter sea ice concentration, wind and geopotential height is used for validation. It has been shown for stable water isotope records from Antarctic ice cores, that single records contain little climatic information on annual time scales, i.e. that it takes several cores to increase the meaningful time period to interpret (Münch and Laepple, 2018). The authors recommend to average over several cores earlier in the text (line229) but interpret their single records with respect to climatic parameter. Assuming that sodium is deposited with the snow (like stable water isotopes) these findings (and the approaches to solve the issue) could be considered similar – i.e. that it makes sense to look and larger time scales than annual for records from the East Antarctic Plateau.**

While we do agree with this to some extent, many in the community would argue that single ice cores do contain meaningful climatic information. It is not always possible, and not always necessary, to combine data to extract a statistically significant relationship with a climate variable. However, we recognise that for stable water isotopes, and especially records in low accumulation sites, combining records is an advantage.

The guidelines for a data descriptor state that extensive interpretations are outside of the scope of the journal. However, we have included this initial interpretation step because we felt it would be beneficial to future users. We agree that more in-depth analysis and interpretation, including combining record, is required and this is the basis for future publications.

**Line 277: It is not explained well, why the correlation analysis includes wind/ geopotential height. What do you expect and why? Is your expectation the same everywhere in Antarctica?**

In the introduction we reviewed the importance of meteorological conditions in driving chemical deposition in Antarctic. Referencing sites where sodium has been used as a tracer for marine-air mass incursions and atmospheric circulation. For this reason, we have included both winds and geopotential height in our first-pass filter.

We have expanded the paragraph in section 3.1.1 to explain this.

"Based on the published literature (section 1) the deposition of $[Na^+]$ and $[SO_4^{2-}]$ has been linked to changes in sea ice, winds, and atmospheric circulation. Thus, these parameters have been chosen for the initial evaluation step."

**Line 288ff: The authors write, that an "interpretation team" is evaluating the results. The objective of this approach is to provide a first level filter for the database. It implies, that the expert's opinion is counted very high, whereas the statistical evidence is not relevant. On what basis did the experts decide? Has there been an objective measure? Is this finding reproducible by others? Maybe it is possible to mention some of the measures taken by the interpretation team here.**

We wanted to avoid over-interpreting the data, but it is important that we don't encourage inappropriate statistical interpretation to impact how the data is used. The use of the interpretation team is to sense check the statistical relationships, not suggest the statistics is irrelevant.

We have tried to demonstrate this is figure 2, by displaying an example of a statistically significant corelation which can not be corroborated with known transport mechanisms. In the chosen example there is a statistically significant corelation with sea ice on the opposite site of the continent to the ice core. While this might be indicating a teleconnection, the direct link between this region and the

chosen ice core site was considered unlikely. We are not suggesting that all users have to follow this method, but we felt it might be useful for those who are less familiar with the data.

We don't have an objective measure of how our decisions are made, but this was the focus of many lengthy discussions. The varying atmospheric processes influencing sites across Antarctica made applying a fixed set of criteria (e.g., distance from source) very difficult. Instead, we have applied a qualitative measure to help guide users, whilst not limiting the future use of the data.

We do identify this method as a potential limitation in section 5.2. Here we identify that 8% of the records were classified as uncertain.

**Line 308ff: In previous studies it is shown, that climatic fields inherit patterns and correlations which lead to p>0.05 probabilities by chance (see Livezey, R. E., & Chen, W. Y., 1983). The presented results have to be taken with care – i.e. to be redone accordingly and/or require a more in-depth discussion.**

Thank you for the reference. Yes, we agree on this and have therefore been cautious not to assume that a statistical correlation proves a climatic connection. We have added an additional sentence in section 4 to strengthen our rational for using the interpretation team.

"Indeed, studies have shown that climatic fields inherit patterns and correlations which can result in statistically significant correlations by chance (Livezey and Chen, 1983)."

**Overall, the interpretation is the weakest part. Maybe it makes more sense to work on the interpretation in more detail in an extra paper and leave it out here.**

We agree that the interpretation is not as extensive as we would expect from a scientific article, however, this was submitted as a data descriptor. And under the journal guidance interpretation cannot be an extensive part of the paper. The scientific interpretation of this data will be presented in more detail in future publications.

References:

Münch, T. and Laepple, T.: What climate signal is contained in decadal- to centennial-scale isotope variations from Antarctic ice cores?, Clim. Past, 14, 2053–2070, https://doi.org/10.5194/cp-14-2053-2018, 2018.

Livezey, R. E., & Chen, W. Y. (1983). Statistical Field Significance and its Determination by Monte Carlo Techniques. Monthly Weather Review, 111(1), 46–59. https://doi.org/10.1175/1520-0493(1983)111<0046:SFSAID>2.0.CO;2

---

## Author Response (AR2)

Dear Dr Heil

Thank you for the additional review and suggestions. Each point is addressed below (in red) and the changes have been made to the revised manuscript.

Comments:

l198: Change of "Flux (ppb kg m -2 ) = concentration (ppb) x snow accumulation (kg m -2 )"

to be a proper equation (with number).

Added two equations with numbers.

We also added proper equations for the calculation of xs SO4 (section 2.6).

l463: Please soften the statement "where a clear mechanism was evident." -- This is agnostic to the reservation expressed by reviewer #2. As noted in your text above (l305ff) "the "interpretation team) to establish if the correlations observed can be attributed to a realistic source region and transport mechanism." No verification is provided here, so the statement "where a clear mechanism was evident" is not justified.

I encourage the authors to revisit their approach of an expert interpretation team.

The description of the data interpretation step has now been expanded in section 4.1. We have removed reference to the interpretation team, as this appears to have caused some confusion.

We also include the statement "based on the parameters applied for this first pass filter" to the text and figure caption. The intention is to acknowledge again that this is not an exhaustive approach, but rather an initial assessment to support the data usage.

In the revised text we have provided examples of the characteristics for a positive case, which we now describe as having a "plausible" transport mechanism or source region. The following text has been amended.

"For sites to be identified as having a relationship with either SIC, atmospheric pressure (z500) or winds (u850 or v850), they had to be supported by a plausible transport mechanism or source region. Therefore, each record was individually evaluated. Sites with a plausible connection were marked as "yes", while sites which did not have a plausible mechanism were marked as "no". In the case of the Ferrigno ice core (Fig. 2a), [Na$^+$] is significantly correlated with SIC is in the adjacent ocean (Amundsen-Ross Sea), and with low pressure anomalies and winds over in the Ross Sea which transport air-masses in a clockwise direction from the source region to the ice core site. Thus, for Ferrigno a plausible source region and transport mechanism has been identified. Conversely, Na at the DFS10 site is also correlated with SIC in the Ross Sea, despite the ice core being located on the opposite side of the continent (Fig. 2b). However, DSF10 [Na$^+$] is not significant correlated with either atmospheric pressure or winds that could transport [Na$^+$] from the Ross Sea to the ice core location. Thus, for DFS10 a plausible source region and transport mechanism has not been identified.

Tab3: Need to define "uncertain" and its use there.

Sites where the transport mechanism was not clear were listed as "uncertain", for example the TA192A ice core (Fig. 2c). Despite the significant corelation between TA192A [Na$^+$] and SIC in the adjacent ocean, the corelations with atmospheric pressure and winds suggest transport that [Na$^+$] from this source region would be transported away from the ice core site. Therefore, it is not possible to identify a plausible source region and transport mechanism for the TA192A site based on the parameters applied for this first pass filter. "

Minor comments:

l97: Need to be consistent "time scales" vs 3 x "timescales" (l 77, 269 and 492)

all converted to timescales.

l116: Correct "O'brien" to "O'Brien".

corrected

l154: Correct "(i.e." to "(i.e.,".

corrected

l159: Correct spelling of "east Antarctica" to capitalize "East Antarctica".

corrected

l182: Correct "e.g." to "e.g.,".

corrected

l209: Correct "(e.g." to "(e.g.,".

corrected

l217: Correct "O'brien" to "O'Brien".

corrected

l252: Ensure that closing bracket "]" is part of the expression "[SO_4^2^-".

Checked all brackets included

l255: Capitalize "table S1" to read "Table S1".

corrected

l268: Change "in the central East Antarctic plateau" to read "on the central East Antarctic plateau".

corrected

l327ff: The explanation of local vs far-field sea-ice information within the ice-core record needs cleaning up to promote the information held within the CLIVASH2k record. Pls detangle and strengthen your argument.

This argument has been updated in terms of the expected elevation of air-parcels reaching the site, which is different at low and high elevations. The work of Suzuki et al., 2013, demonstrate that air-parcel origin height and residence time is much lower at the coast than the interior.

New text added.

We have not applied a uniform cut-off size for the area of correlation or specified a minimum or maximum distance from the source region, as these features will be site specific. For example, the typical air-parcel origin height and residence time over the ice sheet is related to the site topography. As such, air parcels reaching low elevation coastal sites will originate from low elevation sources (e.g., < 2000 m) and have short residence times over the ice sheet (< 20 hours) (Suzuki et al., 2013). Some coastal sites (e.g., Sherman Island) may also be influenced by local orography (mountains), which block air-mass transport and limit the geographical extent of the [$Na^+$] or [$SO_4^{2-}$] source region e.g., Tetzner et al., 2021a. Conversely, air-parcels reaching central Antarctic sites (e.g., South Pole) may originate from elevations in excess of 4000 m, and reside over the ice sheet for more than 120 hours (Suzuki et al., 2013). Thus, higher elevation sites might be influenced by long-range air-mass transport and capture changes in sea ice from relatively distant source regions e.g., Winski et al., 2021.

l329: "Long-range air-masses" are not a thing. Assume you mean "Long-range air-mass transport".

Corrected

l330: Add a comma before "e.g.,".

Corrected

l330: Remove brackets around "(Winski et al., 2021)".

Corrected

l334: Capitalize "figures".

Corrected

l346: Correct "Fifty-Six" to "Fifty six" or change "56".

Corrected

l348: Comment from Reviewer 1 has not been corrected even though ticked off: "Please adjust them according to the values listed in Table 3 (if I understand correctly, it should be 86% compared to 78%)."

--> Pls correct, or explain in the author response.

Corrected – apologies this was missed previously.

l452: Change "the defined criteria." to "the criteria defined here".

Updated

l471: Change "first attempt to compile" to "first compilation of".

Updated

l473: Remove "In this study".

Updated

l475: Change "(if available) " to "(where available)".

Updated

l493: Add "distribution" to read "sea-ice distribution".

Updated